# The Role of the Low-Density Lipoprotein/High-Density Lipoprotein Cholesterol Ratio as an Atherogenic Risk Factor in Young Adults with Ischemic Stroke: A Case—Control Study

**DOI:** 10.3390/brainsci13081180

**Published:** 2023-08-09

**Authors:** Sibel Ciplak, Ahmet Adiguzel, Yusuf Ziya Deniz, Melike Aba, Unal Ozturk

**Affiliations:** 1Department of Neurology, Turgut Ozal University Faculty of Medicine, Malatya 44090, Turkey; 2Department of Neurology, Inonu University Faculty of Medicine, Malatya 44210, Turkey; 3Department of Neurology, Siirt Education and Research Hospital, Siirt 56000, Turkey; 4Department of Neurology, Mehmet Akif Inan Education and Research Hospital, Sanliurfa 63330, Turkey; 5Department of Neurology, University of Health Sciences, Gazi Yasargil Education and Research Hospital, Diyarbakir 21070, Turkey; drunalozturk@gmail.com

**Keywords:** ischemic stroke, LDL-C/HDL-C, atherosclerosis, hyperlipidemia, cholesterol

## Abstract

Dyslipidemia is a major atherogenic risk factor for ischemic stroke. Stroke patients tend to have high levels of total cholesterol (TC) and low-density lipoprotein cholesterol (LDL-C) and low levels of high-density lipoprotein cholesterol (HDL-C). Therefore, it is noteworthy that there has been an increase in ischemic stroke cases in young and elderly individuals in recent years. This study investigated the TC/HDL-C ratio and the LDL-C/HDL-C ratio, which may be more specific and common lipid parameters in young patients with ischemic stroke. This study aimed to demonstrate the sensitivity and specificity of TC/HDL-C and LDL-C/HDL-C ratios as atherogenic markers for young adult ischemic strokes. This trial was conducted as a retrospective case—control study. A total of 123 patients (patient group) and 86 healthy individuals (control group) aged 18–50 years were randomly selected from four different hospitals. Lipid parameters and TC/HDL-C and LDL-C/HDL-C ratios were compared between these two groups. The mean age was 38.8 ± 7.3 years in patients and 37.7 ± 9 years in controls (*p* > 005). The HDL-C levels were 39.1 ± 10.8 mg/dL in patients and 48.4 ± 13.8 mg/dL in controls (*p* < 0.001). LDL-C/HDL-C ratios were 3.23 ± 1.74 and 2.38 ± 0.87, and TC/HDL-C ratios were 5.24 ± 2.31 and 4.10 ± 1.25 in the patient and control groups, respectively (*p* < 0.001). The LDL-C/HDL-C and TC/HDL-C cutoff values in ROC analyses were 2.61 and 4.40 respectively; the AUCs (95% CI) were determined to be 0.680 (0.608–0.753) and 0.683 (0.610–0.755) (*p* < 0.001), respectively. An increased risk of stroke was observed in those with a high LDL-C/HDL-C ratio (OR = 1.827; 95% CI = 1.341–2.488; *p* < 0.001). Our study obtained similar results when we compared the mean TC and LDL-C levels between the two groups. However, considering the TC/HDL-C and LDL-C/HDL-C ratios, it is noteworthy that there was a significant difference between the patient and control groups.

## 1. Introduction

Young adult ischemic stroke is a neurovascular disease that presents at ages 18–50. It causes loss of function and mortality as a result of neuronal damage due to insufficient cerebral blood flow. In recent years, several studies have reported an increase in stroke rates among young people [1]. Ischemic strokes in young adults account for 10–15% of all ischemic strokes [2]. Stroke incidence in young adults increases with age, especially in those over 35 years. Ischemic stroke is more common in males than in females [3]. It is a major public health problem worldwide. Its incidence is increasing and it causes high morbidity and mortality [4,5]. Smoking, diabetes mellitus (DM), hypertension (HT) and hypercholesterolemia are the most common etiological causes of ischemic stroke in young adults [6]. To prevent and manage strokes, it is therefore important to identify the cause of stroke. Atherosclerosis and cardioembolic disease remain important risk factors. However, the cause of 1/3 of strokes is still unknown [7]. Atherosclerosis causes vascular wall lesions due to the accumulation of cholesterol-rich lipids (LDL-C, low-density lipoprotein cholesterol and VLDL-C, very low-density lipoprotein cholesterol) in the arterial intima [8]. It is well known that there is a specific relationship between atherogenic lipid disorders and symptomatic intracranial atherosclerotic disease [9]. There is a straight-line association between serum total cholesterol (TC), triglyceride (TG) and LDL-C levels and the risk of stroke. The literature has shown that there is a substantial increase in all cholesterol levels except HDL-C (high-density lipoprotein cholesterol) in young adult ischemic stroke patients, regardless of the underlying etiology, as in other stroke patients [10]. In most studies, dyslipidemia was defined as either high LDL-C or low HDL-C, but the association between different lipid variables and stroke was not investigated. Therefore, it is important to investigate the secondary parameters of lipid profiles and their association with stroke in young patients [4].

The primary objective of this study was to investigate lipid parameters in young patients with ischemic stroke. Furthermore, the LDL-C/HDL-C and TC/HDL-C ratios were computed and then compared with the control group. Our primary goal in this article is to demonstrate that LDL-C/HDL-C and TC/HDL-C ratios have more reliable results as atherogenic risk factors. We believe this paper will add to the knowledge, especially as there is no consensus in the literature regarding LDL-C/HDL-C and TC/HDL-C ratios in young stroke patients [10,11].

## 2. Materials and Methods

### 2.1. Data and Variables

This trial was conducted as a retrospective case—control study. It included four hospitals located in different cities in eastern Turkey. The clinical research ethics committee approved the study with decision number 2022/36. The study included 123 patients with ischemic stroke and 86 healthy controls. The patient group consisted of individuals who registered at the hospital with a stroke between 2018 and 2021. The control group was randomly selected from healthy individuals. The diagnosis of ischemic stroke was checked by two neurologists. The inclusion criteria were as follows: 18–50 years of age, radiological findings compatible with ischemic stroke using computed tomography (CT) and diffusion-weighted magnetic resonance imaging (DW-MRI), and clinical diagnosis of stroke according to AHA/ASA guidelines [12]. The exclusion criteria were recurrent stroke, family history of hereditary hyperlipidemia, use of lipid-lowering medication, and the presence of diagnoses affecting lipid parameters (malignancy, chronic infection, obesity, vegetarianism, pregnancy, postpartum, lactation, etc.). Eight patients without a diffusion MRI scan and twelve patients with missing lipid parameters were excluded from the study.

The participants’ demographic characteristics, such as age, sex, background, family history and habits (smoking, alcohol), were recorded. To minimize stress-induced hyperlipidemia in the acute phase, lipid parameters, thyroid function test panels and vitamin B12 and HbA1c tests were assessed in peripheral venous blood samples obtained after 12 h of fasting on the third day of stroke. Samples were collected in EDTA blood collection tubes, and lipid parameters and other blood tests were assessed. Abbott Architect^®^ C16000 and Beckman Coulter^®^ AU 5800 analyzers were used for the analyses. The reference values and units of measurement used were the same in all four centers where the study was carried out. For the lipid parameters analyzed, LDL-C was calculated using the following formula [LDL-C = TC-(HDL-C + TG/5)]. Normal levels were defined as TC < 200 mg/dL, LDL-C < 130 mg/dL, HDL-C > 40 mg/dL and TG < 150 mg/dL. The lipid parameters LDL-C to HDL-C and TC to HDL-C ratios were calculated.

### 2.2. Statistical Analysis

Statistical analysis was performed using SPSS^®^ for Windows, version 26.0 software (IBM Corp., Armonk, NY, USA). The Kolmogorov-Smirnov test was performed to determine the normality of the data. Frequency and descriptive tests were used for the data of dependent variables. The chi-square test was used to compare categorical variables. The Mann-Whitney U test was used for data that were not normally distributed. Binary logistic regression analysis was performed. *p* < 0.05 was considered statistically significant.

## 3. Results

### 3.1. Descriptive Analysis

In this trial, the number of participants in the patient group was 123 (m/f: 54 (43.9%)/69 (56.1%)), and the control group had 86 (m/f: 45 (52.3%)/41 (47.7); *p* > 0.05). The mean ages of the patients and controls were 38.8 ± 7.3 (min–max: 19–50) and 37.7 ± 9 (min–max: 20–50) years, respectively. Comparing only the TC, LDL-C and VLDL-C levels between the two groups, we see that they had similar results. The HDL-C levels in the patients and controls were 39.1 ± 10.8 mg/dL and 48.4 ± 13.8 mg/dL, respectively (*p* < 0.001). To assess the LDL-C/HDL-C ratio and the TC/HDL-C ratio, the main focus of this study, we analyzed the significant differences between the two groups. The LDL-C/HDL-C levels in patients and controls were 3.23 ± 1.74 and 2.38 ± 0.87, respectively (*p* < 0.001). The study showed the importance of considering lipid parameters and the HDL-C ratio as atherogenic risk factors in young stroke patients. Triglyceride was higher in the patient group (200 ± 125 mg/dL) than in the control group (*p* < 0.05). Significant discrepancies were found between the two groups in lipids and other key laboratory parameters (Table 1).

The LDL-C/HDL-C and TC/HDL-C cutoff values in ROC analyses were 2.61 and 4.40, respectively. In addition, the area under the curves (AUCs) were (95% CI) 0.680 (0.608–0.753) and 0.683 (0.610–0.755), respectively (*p* < 0.001) (Figure 1).

Major risk factors for ischemic stroke were compared between the patient and control groups. The most common risk factors in the patient group were defined as HT (*n* = 47), DM (*n* = 32), coronary artery disease (CAD) (*n* = 23), and atrial fibrillation (AF) (*n* = 14). The prevalence of smoking habits in patients and controls was 45% (*n* = 55) and 11% (*n* = 9), respectively (*p* < 0.001). One of the most remarkable results of this study was the significantly lower HDL-C levels in the patient group. TC/HDL-C and LDL-C/HDL-C ratios were significantly related to an increased risk of stroke in univariate regression analyses (*p* < 0.001) (Table 2).

### 3.2. AUC/Logistic Regression Results

The rates of HDL-C < 50 mg/dL in the patient and control groups were 84% (*n* = 103) and 55% (*n* = 64), respectively (*p* < 0.001). The LDL-C >130 mg/dL rate was 37% (*n* = 46) in patients and 24% (*n* = 21) in controls (*p* < 0.05) (Figure 2). These two results may suggest that low HDL-C is a more acceptable risk factor than high LDL-C in young ischemic stroke patients. The distributions of TC/HDL-C and LDL-C/HDL-C ratios were plotted by age group (Figure 3).

## 4. Discussion

### 4.1. Summary and Contributions

Transendothelial transport of LDL induced by focal hemodynamic changes plays a key role in the formation of atherosclerotic plaque [13]. Meanwhile, HDL plays a protective role against intracranial atherosclerosis [14]. In intracranial arteries, the local geometry of arteries and stents can significantly influence the LDL filtration rate. The roles of anatomic/geometric variation, hemodynamic parameters and dyslipidemia, as well as their interaction, in the development of intracranial artery disease and stroke deserve further exploration [15]. This study examined the TC/HDL-C ratio, known to be an atherogenic risk factor in coronary heart disease for young ischemic stroke patients, due to similar pathophysiological mechanisms. In this study, the LDL-C/HDL-C and TC/HDL-C ratios were more significant than the LDL-C, HDL-C and TG parameters in the lipid panel in the group of young adult ischemic patients. The findings included higher LDL-C/HDL-C and TC/HDL-C ratios in young ischemic stroke patients. Based on studies in the literature, the TC/HDL ratio could be used as a routine laboratory parameter [16]. Furthermore, studies have shown that total cholesterol, as well as LDL-C/HDL-C and TG/HDL-C ratios, increase the risk of cardiovascular disease [10,17]. Therefore, the high LDL-C/HDL-C ratio in young ischemic stroke patients in our study is considered a valuable finding that may guide other similar studies in the future.

At least one of hypercholesterolemia, smoking or hypertension can be considered as a modifiable vascular risk factor in most common of young ischemic stroke patients. [18,19]. Cholesterol is an essential component of cell membranes and a precursor of steroid hormones. The primary lipoproteins that carry cholesterol are LDL-C and HDL-C. These lipoproteins are crucial for the transport of neuronal signaling molecules [20]. Elevated triglyceride levels and lower HDL-C levels are important risk factors for coronary disease and ischemic stroke [21,22,23,24]. T. Sun et al. emphasized that the LDL-C/HDL-C ratio is more valuable than LDL-C and HDL-C levels alone in lipid panel analysis in determining the severity of coronary artery disease. Their study retrospectively analyzed data from 1351 cases of myocardial ischemia diagnosed between 2018 and 2019. When comparing patients and healthy controls, the LDL/HDL ratio was higher in patients with CAD (2.94 ± 1.06 vs. 2.36 ± 0.78, *p* < 0.05). When comparing patients and healthy controls, the LDL/HDL ratio was higher in patients with CAD (2.94 ± 1.06 vs. 2.36 ± 0.78, *p* < 0.05) [25]. Zhuchao et al. investigated the role of the LDL-C/HDL-C ratio on carotid plaque formation. The researchers screened 2191 participants and noted that carotid plaque formation increased as the LDL-C/HDL-C ratio increased, particularly in people with diabetes and dyslipidemia [26]. These studies are evidence that the LDL-C/HDL-C ratio is an indicator of atherosclerotic plaque formation. The LDL-C/HDL-C ratio is a factor that predisposes young patients to stroke, although vascular stenosis was not assessed in our study. Another similar study examined whether LDL-C/HDL-C ratios were superior to LDL-C or HDL-C levels in predicting disease severity in CAD. Its participants were divided into two groups according to the LDL-C/HDL-C threshold. The CAD incidence was higher in those with LDL-C/HDL-C > 2.517 (*p* < 0.05) [25].

Our study found that the optimal cut-off value for LDL-C/HDL-C to predict the risk of ischemic stroke in young patients was 2.615. The cutoff value for TC/HDL-C was 4.405. All these data show that TC, LDL-C and HDL-C are more significant independent markers for young ischemic stroke when considering TC/HDL-C and LDL-C/HDL-C, rather than considering them separately. In this way, it will be possible to achieve more successful outcomes with effective treatment early in the follow-up of young ischemic stroke patients, the number of which is increasing. Dyslipidemia was reported as an independent risk factor in a study that included 3944 young people suffering from ischemic stroke in 12 countries in 2012 [27]. Likewise, in a study by Renna et al., which examined risk factors in 150 young ischemic stroke patients, dyslipidemia (52.7%) was found to be the most common risk factor [28]. Another trial also reported that dyslipidemia was the second most common risk factor (41.1%) in a group of 253 young people suffering from ischemic strokes [29]. Although it is widely known that dysregulation of lipid metabolism causes stroke, our study highlighted its importance in young people. According to Grobbe et al., the risk of stroke increases by approximately ¼ for every 1 mmol increase in TC. In addition, when the TC level is above 280 md/dL, the risk of stroke doubles [30]. Likewise, in the present study, TC levels were higher in patients than they were in controls. Therefore, we found an increased risk of stroke in those with a high LDL-C/HDL-C ratio (OR = 1.827; 95% CI = 1.341–2.488; *p* < 0.001). Likewise, in regression analyses we observed an increase in the risk of stroke with an elevated TC/HDL-C ratio (*p* < 0.001). In another study published in 2018, 112 young ischemic stroke patients and 113 healthy young adults were compared in terms of LDL, HDL, TG, TC levels and LDL/HDL, TG/HDL and TC/HDL ratios. Although all lipid levels were higher in young ischemic stroke patients, only TG levels and TG/HDL ratios were statistically significant (*p* < 0.05) [10]. In our trial, although TG, TC and LDL-C values were numerically higher, TC/HDL and LDL/HDL ratios were higher in young patients than they were in elderly patients (*p* < 0.05). Therefore, considering TC/HDL and LDL-C/HDL-C ratios, rather than relying on the lipid panel, may be more informative.

Zhao et al. assessed 1111 people aged <65 years and 963 people aged ≥65 years for lipid profiles. Regardless of sex, TC and TG levels between the ages of 40 and 69, as well as LDL levels between the ages of 50 and 69, were significantly higher than those of individuals aged 80 and above [11]. Although the total lipid panel of young ischemic stroke patients was higher than that of older patients, our study highlighted that the TC/HDL and LDL/HDL ratios were higher in all young ischemic stroke patients independent of gender. Therefore, examining only TC, LDL-C and HDL-C levels as indicators of atherosclerosis was not considered sufficient.

### 4.2. Limitations

Ischemic stroke in young adults is an important area of research. It is important to note that we limited this study. The relationship between the etiology of stroke and lipid ratios and their association with stroke severity has not been evaluated. In our view, this is an important issue that should be the subject of a separate study. Apolipoprotein A-B testing, which can provide a better understanding of the effect of lipid metabolism on stroke, could not be carried out in this study due to a lack of funding.

## 5. Conclusions

We observed that individual TG, TC and LDL-C levels were numerically higher in young ischemic stroke patients than in controls. We found statistically significant differences between the two groups when we examined the TC/HDL-C and LDL-C/HDL-C ratios. More comprehensive and comparable studies with larger populations are needed to determine the future accepted cutoff values for these two parameters.

## Figures and Tables

**Figure 1 brainsci-13-01180-f001:**
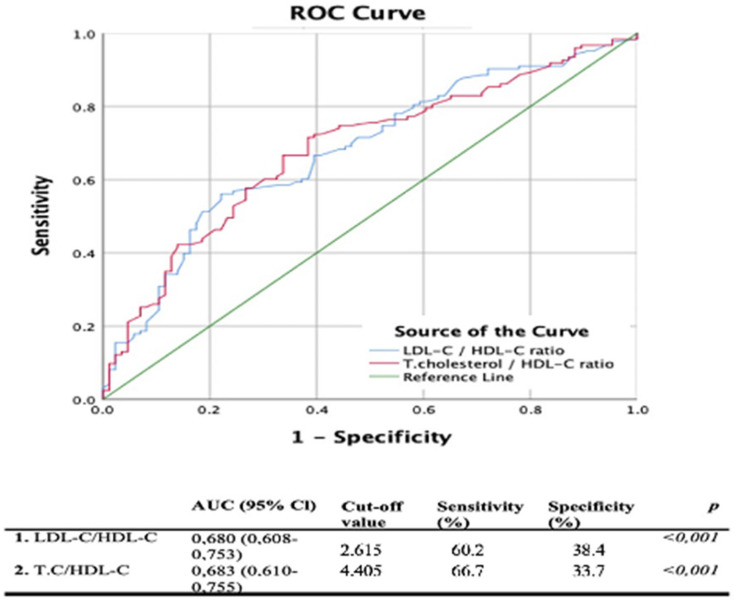
ROC curve analysis for TC/HDL-C and LDL-C/HDL-C in young adults with strokes.

**Figure 2 brainsci-13-01180-f002:**
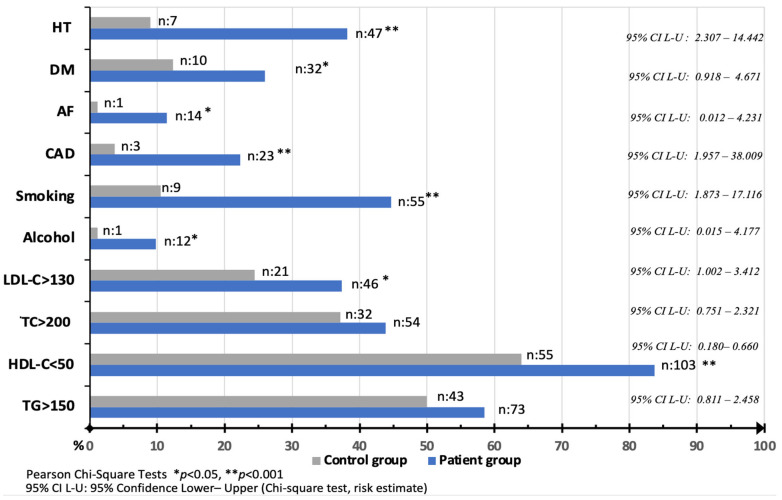
Comparison of major risk factors between the two groups. X-axis indicates percentages.

**Figure 3 brainsci-13-01180-f003:**
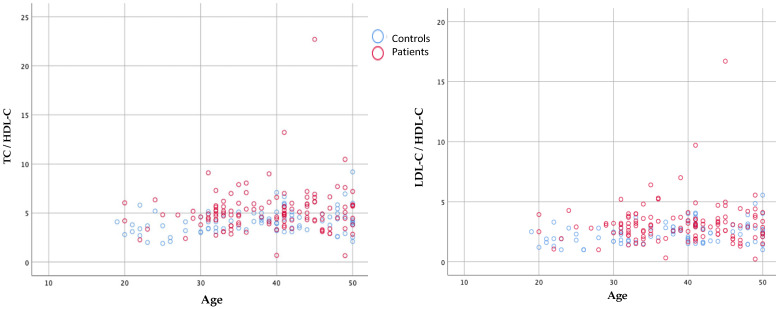
Frequency distribution of TC/HDL-C and LDL-C/HDL-C ratios by age.

**Table 1 brainsci-13-01180-t001:** Analysis of lipid levels and other key laboratory parameters.

	Patients (*n* = 123)		Controls (*n* = 86)			
	Mean ± SD	Median (Min–Max)	Mean ± SD	Median (Min–Max)	Test Statistics	*p*
**TC**	196.3 ± 49.6	192 (120–386)	188.1 ± 34.6	187 (116–270)	5219.5	0.87
**LDL-C**	119.1 ± 39.4	113.3 (57–285)	109.6 ± 29.4	113.8 (55.5–174.2)	4898	0.36
**HDL-C**	39.1 ± 10.8	37.2 (17–89)	48.4 ± 13.8	45.7 (24–94.5)	3106	**0.001 ***
**VLDL-C**	38.5 ± 21.7	34 (12–98)	34.5 ± 17.9	31.6 (8.4–96.8)	3138.5	0.38
**TG**	200 ± 125.1	166 (52–704)	151.1 ± 101.5	130 (29–487)	4245.5	**0.01**
**TC/HDL-C**	5.24 ± 2.31	4.90 (0.7–21.2)	4.10 ± 1.25	4.10 (1.9–9.2)	3355.5	**0.001 ***
**LDL/HDL-C**	3.23 ± 1.74	3.02 (0.3–15.8)	2.38 ± 0.87	2.40 (1.0–5.6)	3380	**0.001 ***
**B12 vit.**	318.1 ± 136.7	290 (50–941)	325 ± 127.4	292.3 (149.5–812.5)	4347.5	0.128
**TSH**	1.77 ± 1.25	1.4 (0.1–7.6)	2.16 ± 1.53	1.8 (0.3–8.4)	4232	0.09
**T4**	5.14 ± 6.16	1.2 (0.2–20.1)	6.77 ± 7.02	1.4 (0.8–20.7)	4090	0.21
**HbA1c%**	6.19 ± 1.49	5.7 (4.5–11.9)	5.67 ± 1.34	5.5 (4.1–12.5)	2669.5	**0.005**

Mann-Whitney U Test * *p* < 0.001. SD: standard deviation; TC mg/dL: total cholesterol; LDL-C mg/dL: low-density lipoprotein cholesterol; HDL-C mg/dL: high-density lipoprotein cholesterol; VLDL-C mg/dL: very low-density lipoprotein cholesterol; TG mg/dL: triglyceride; TSH mIU/L: thyroid stimulating hormone; T4 mIU/L: thyroxine; HbA1c%: glycolized hemoglobin A1c.

**Table 2 brainsci-13-01180-t002:** Predictive value of lipid parameters regarding ischemic stroke in young adults.

	Multivariate *		Univariate	
	OR (95% CI)	*p*	OR (95% CI)	*p*
**TC**	1.007 (0.987–1.027)	0.499	1.007 (0.996–1.008)	0.535
**LDL–C**	0.992 (0.977–1.008)	0.325	1.001 (0.995–1.007)	0.751
**HDL–C**	0.949 (0.891–1.011)	0.103	0.942 (0.918–0.966)	**<0.001**
**TG**	1 (0.996–1.004)	0.904	1.003 (1.000–1.006)	0.026
**TC/HDL–C**	0.954 (0.533–1.708)	0.874	1.541 (1.232–1.928)	**<0.001**
**LDL–C/HDL–C**	1.405 (0.706–2.793)	0.333	1.827 (1.341–2.488)	**<0.001**
**Constant**	4.89 (0–0)	0.275		

* Cox and Snell R Square = 0.138; Nagelkerke R Square = 0.186. OR: odds ratio. CI: confidence interval; TC mg/dL: total cholesterol; LDL-C mg/dL: low-density lipoprotein cholesterol; HDL-C mg/dL: high-density lipoprotein cholesterol; TG mg/dL: triglyceride.

## Data Availability

The complete de-identified dataset is available from the corresponding author on reasonable request.

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
