# Peer review of "The Role of the Low-Density Lipoprotein/High-Density Lipoprotein Cholesterol Ratio as an Atherogenic Risk Factor in Young Adults with Ischemic Stroke: A Case—Control Study"

_brainsci, 2023, doi:10.3390/brainsci13081180_

Round 1

Reviewer 1 Report

see below

Author Response

First of all, I would like to thank you for your positive feedback, helpful suggestions and critical comments. I have tried to respond to your suggestions as seriously as possible.

The following is a list of the revisions that were made.

  1. The requested explanations about the LDL-C/HDL-C ratio have been added to the discussion section.

‘We can also speculate that the LDL …’

  1. We have included our suggestions and comments on the TC/HDL-C ratio in the discussion section.

              ‘The TC/HDL-C ratio was found to be higher than the …’

  1. The recommended article has been used as a reference.

‘Ciarambino T, Crispino P, Mastrolorenzo E,…’

4. The article has been proofread and edited by a native speaker. The certificate has been uploaded.

Reviewer 2 Report

The authors investigated the lipid ratios in young patients with ischemic stroke in comparison with normal controls. Overall, the experiment is well designed. However, the is a need for more comprehensive introduction on background and in-depth discussion from a pathophysiological view.

1. In the introduction, it is essential to highlight the research gap of this study. Dyslipidaemia can coexist with cardiovascular diseases and many neurological dysfunctions, e.g., dizziness, as observed in old patients (Refer: 10.1007/s00415-021-10899-7, 10.15420/ecr.2019.06). However, dyslipidaemia in young adults is not significantly associated with the risk of all-cause stroke. One possible explanation is that dyslipidaemia might not be a leading risk factor compared with old patients, for whom dyslipidaemia would increase the risk of stroke due to large artery disease or small vessel disease. Another explanation might be that most studies defined dyslipidaemia as either a high low-density lipoprotein cholesterol or low high-density lipoprotein cholesterol, but did not investigate the association between different lipid variables and stroke (Refer: 10.1136/jnnp-2019-322424). Therefore, it is essential to investigate the secondary parameters of lipid profiles and their association with stroke in young patients.

2. In Table 1, I would like to recommend using <0.0001 instead of 0.0000.

3. Please enlarge the fonts in Figures 1 and 2 and make sure the details can be clearly shown on A4 size.

4. The discussion on the underlying pathophysiology is necessary. Transendothelial transport of LDL induced by focal hemodynamic changes plays a key role in the formation of atherosclerotic plaque (Refer: 10.1088/2057-1976/aa9a09). In intracranial arteries, local geometry of artery and stent can significantly influence the LDL filtration rate (Refer: 10.3389/fneur.2022.1067566). Meanwhile, HDL plays a protective role against intracranial atherosclerosis (Refer: 10.3389/fneur.2020.504219). Therefore, dyslipidaemia is a risk factor of not only ischemic stroke and also its recurrency (10.1136/svn-2022-001606). The roles of anatomic/geometric variation,  hemodynamic parameters, and dyslipidaemia, as well as their interaction, in the development of intracranial artery disease and stroke deserve further exploration.

Overall it is well written. Some details need double check and further improvement.

Author Response

First of all, I would like to thank you for your positive feedback, helpful suggestions and critical comments. I have tried to respond to your suggestions as seriously as possible.

The following is a list of the revisions that were made.

  1. The introduction section has been made more informative by taking into account your suggestions. The suggested article has been examined ( Refer: 10.1136/jnnp-2019-322424 ).   
  2. Table 1 has been revised (revised as p>0.001)
  3. Dimensions of Figures 1 and 2 rearranged.
  4. In the discussion section, pathophysiological mechanisms were mentioned.The suggested articles has been examined and indexed.
  5. The article has been proofread and edited by a native speaker. The certificate has been uploaded.

Reviewer 3 Report

Dear Authors; I found your case control study work on investigation of role of the LDL/HDL cholesterol ratio as an atherogenic risk factor in young adults with ischemic stroke very interesting.  It needs some extra work prior to processing it further. Regards. P.S. 

[1] Writing: 

1-1 Missing Abbreviation List: Add these at the end of text right before references section for your readers referral. Example: Abbreviations: TC: Total Cholestrol; ....

1-2 Missing Citations: 

This sentence claimed in lines 62-65 needs several citations !

"We believe this paper will add to the knowledge, especially as there is no consensus in the literature regarding LDL-C/HDL-C and TC/HDL-C ratios in young stroke patients."

1-3 Missing Subsections in "2.MAterials & Methods": It is written substandard . Break it down with subsections: 2.1 Data & Variables; 2.2. Statistical Analysis.

1-4 Missing Geographical Area Map: Add the map of Turkey with 4 cities in it showing your study location to the readers. This will help  your work citation  in the future for meta analysis !

1-5 Missing Subsection in "3. Results". Again, this is written substandard. Add subsections:  3.1 Descriptive analysis 3.2. AUC/Logistic Regression Results

1-6 Discussion: It is very hard to follow it up. Break it down to subsections like this: 4.1. Summary & Contributions; 4.2. Strengths & Limitations; 4.3. Future Work

[2] Statistical: 

2-1  Missing Matching Process: In clinical trials including case-control studies the both arms need to be as far as possible similar except disease status. Which strategy you used ? Stratification ? Propensity score ? what ? Note that without clear clarification your study design from statistical point of view in unwarranted and unpublishable.

2-2 Missing Consort Diagram:  Add this to show readers how selection process was performed for each arm of the study.

2-3 Missing analysis Package Citations: Add these for Lines 86, 87. Statisticians readers need to verify them !

2-4 Missing Logistic Regression Table: Add this in lines 130-131. 

2-5 Figure  2: The plot is so substandard and disappointing. Missing 95%CI intervals on top of each bar. You need to add these to tell the readers the span of estimates. 

2-6 Missing Figure 3: Need to add the trend of estimates per trial arm over age span to tell the readers-specially clinicians- what is going on there ! Plot  something lines this:  X axis: age, Y Axis: n_outcome   Legend:  Two arms: Case/Control. There are potentially other interesting trend. Please add them wherever deem adding more insights to your work. 

It needs some minor editing. 

Author Response

First of all, I would like to thank you for your positive feedback, helpful suggestions and critical comments. I have tried to respond to your suggestions as seriously as possible.

The following is a list of the revisions that were made.

Best regards

  • There is no section in the journal's manuscript template for the list of abbreviations at the end of the article. Therefore, abbreviations are explained where necessary.
  • The missing citation has been updated. The addition of a reference to the sentence mentioned was overlooked.
  • Subsection added to the Methods section.
  • The cities participating in the study are Malatya, Sanliurfa, Diyarbakir and Siirt. These provinces are in the east of Turkey. If there is a reasonable request, city names can be provided to readers by email.
  • Subsection added to the Results section.
  • Binary logistic regression analysis was performed (Table 2).
  • The procedure for the identification of patients and controls has been revised in the Methods section.

‘The patient group consisted of…’

  • The article has been proofread and edited by a native speaker. The certificate has been uploaded.

Round 2

Reviewer 2 Report

Thanks for the update. My earlier comments have been addressed. 

It is essential to carefully check the format in proofreading.

Author Response

Reviewer comments: Thanks for the update. My earlier comments have been addressed. 

I would like to thank you for your positive feedback.

Best regards

Reviewer 3 Report

Dear Authors, your revision does not satisfy me. Items 1-1, 1-6, 2-5 and 2-6 are critical. Regards.

Round 2

First of all, I would like to thank you for your positive feedback, helpful suggestions and critical comments. The following is a list of the revisions that were made.

Best regards

Reviewer comments: There is no section in the journal's manuscript template for the list of abbreviations at the end of the article. Therefore, abbreviations are explained where necessary.

Revision: List of abbreviations has been added at the end of the article.

Reviewer comments: Discussion: It is very hard to follow it up. Break it down to subsections like this: 4.1. Summary & Contributions; 4.2. Strengths & Limitations; 4.3. Future Work

Revision: The discussion section is divided into subheadings.

Reviewer comments: Figure 2: The plot is so substandard and disappointing. Missing 95%CI intervals on top of each bar. You need to add these to tell the readers the span of estimates

Revision: 95%CI intervals on top of each bar has been added.
